# CEBPA-Regulated Expression of *SOCS1* Suppresses Milk Protein Synthesis through mTOR and JAK2-STAT5 Signaling Pathways in Buffalo Mammary Epithelial Cells

**DOI:** 10.3390/foods12040708

**Published:** 2023-02-06

**Authors:** Xinyang Fan, Lihua Qiu, Wei Zhu, Lige Huang, Xingtiao Tu, Yongwang Miao

**Affiliations:** Faculty of Animal Science and Technology, Yunnan Agricultural University, Kunming 650201, China

**Keywords:** buffalo, *SOCS1*, CEBPA, milk protein synthesis, transcriptional regulation

## Abstract

Milk protein content is a key quality indicator of milk, and therefore elucidating its synthesis mechanism has been the focus of research in recent years. Suppressor of cytokine signaling 1 (*SOCS1*) is an important inhibitor of cytokine signaling pathways that can inhibit milk protein synthesis in mice. However, it remains elusive whether *SOCS1* plays roles in the milk protein synthesis in the buffalo mammary gland. In this study, we found that the mRNA and protein expression levels of *SOCS1* in buffalo mammary tissue during the dry-off period was significantly lower than those during lactation. Overexpression and knockdown experiments of *SOCS1* showed that it influenced the expression and phosphorylation of multiple key factors in the mTOR and JAK2–STAT5 signaling pathways in buffalo mammary epithelial cells (BuMECs). Consistently, intracellular milk protein content was significantly decreased in cells with *SOCS1* overexpression, while it increased significantly in the cells with *SOCS1* knockdown. The CCAAT/enhancer binding protein α (CEBPA) could enhance the mRNA and protein expression of *SOCS1* and its promoter activity in BuMECs, but this effect was eliminated when CEBPA and NF-κB binding sites were deleted. Therefore, CEBPA was determined to promote *SOCS1* transcription via the CEBPA and NF-κB binding sites located in the *SOCS1* promoter. Our data indicate that buffalo *SOCS1* plays a significant role in affecting milk protein synthesis through the mTOR and JAK2-STAT5 signaling pathways, and its expression is directly regulated by CEBPA. These results improve our understanding of the regulation mechanism of buffalo milk protein synthesis.

## 1. Introduction

The synthesis and secretion of milk is the most important function of the mammalian mammary gland, and it is one of the most metabolically demanding stages in mammalian life [1,2]. Milk not only provides the newborn with necessary nutrients, but also with a complex repertoire of agents required for healthy development [3]. As the primary solid components of milk, milk proteins mainly include the caseins and whey proteins, which are synthesized in mammary epithelial cells and mainly by ribosomes in the rough endoplasmic reticulum [4]. Milk proteins contain a large number of essential amino acids, which is essential to meet the nutritional needs of newborns, so milk protein content can serve as a quality indicator for milk [5]. The synthesis of milk protein is regulated by a variety of signaling pathways and hormones at multiple levels. The mammalian target of rapamycin (mTOR) pathway and janus kinase 2/signal transducer and activator of transcription 5 (JAK2–STAT5) pathway are well known to be the crucial signaling pathways involved in regulating milk protein synthesis in the mammary gland [5]. Studies have shown that multiple genes are associated with the regulation of milk protein synthesis in cows, such as *ULF1*, *TWF1* and *NUCKS1* [1,6,7].

The suppressor of cytokine signaling (SOCS) family proteins, the critical inhibitors of cytokine signaling pathways, play an important role in mammary gland development, along with the JAK-STAT pathway [8]. In the mammary gland, the SOCS inhibit the ability of prolactin (PRL) to stimulate milk synthesis [9]. As an important member of this family, *SOCS1* is involved in mammary gland development and a variety of cytokine signal transduction [10]. In mouse mammary epithelial cells, the expression of *SOCS1* can be induced by PRL, suggesting that *SOCS1* is implicated in PRL-stimulated regulation of cell signaling [11]. Meanwhile, the evidence in vivo confirms that *SOCS1* attenuates the prolactin receptor (PRLR) signaling through negative feedback during pregnancy and lactation [9]. A previous study reported that several SNPs close to *SOCS1* are significantly associated with milk protein yield in dairy cattle [8]. In addition, the deletion of *SOCS1* restores milk protein expression in PRLR-deficient mice [10]. These findings reveal that *SOCS1* plays a key role in the regulation of milk protein synthesis.

The CCAAT/enhancer binding proteins (CEBPs) are a family of transcription factors implicated in the growth and differentiation of mammary epithelial cells [12]. They contain highly conserved basic leucine zipper motifs that mediate dimerization and DNA binding at their carboxyl terminus [13]. CEBPs are differentially expressed throughout mammary gland development, and they can bind to the promoter of *CSN2* to regulate its expression [14]. CEBPA, the first member of this family to be identified, is regulated by lactogenic hormones in mammary epithelial cells. Although lactogenic hormones have no effect on CEBPA protein level, they down-regulate the DNA binding activity of CEBPA [12]. In addition, CEBPA is an important lipogenic transcription factor that targets *FASN* and *CD36* related to lipid synthesis [15,16]. In bovine mammary epithelial cells, CEBPA is involved in the regulation of milk fat synthesis [17]. CEBPA was found to directly activate miR-29b expression, while miR-29b induced *SOCS1* expression via promoter demethylation [18]. However, direct regulation of *SOCS1* by CEBPA has not been reported.

At present, buffalo milk production accounts for 13% of the world’s total milk production, and water buffalo has become the second largest source of milk in the world [19,20]. Compared with cow milk, buffalo milk has a higher content of milk protein. Although *SOCS1* has been proven to play essential roles in milk protein synthesis in mice, its functions in milk protein synthesis, and the interaction mechanism with CEBPA in buffalo mammary gland are unclear. We hypothesized that *SOCS1* could regulate the network associated with buffalo milk protein synthesis and that its expression could be regulated by CEBPA. The purpose of this study was to determine the role of *SOCS1* in buffalo milk protein synthesis and to clarify the interaction mechanism between this protein and CEBPA through functional experiments at the cellular level. This study can lay a foundation for elucidating the regulation of buffalo milk protein synthesis.

## 2. Materials and Methods

### 2.1. Animals and Sampling

Eight healthy female Binglangjiang buffalo (river type) aged 4 years old in the same management conditions were selected for tissue sample collection. The samples from the mammary gland, liver, ovary, lungs, rumen, cerebellum, kidney, brain, heart, pituitary, spleen and muscle were collected from 4 buffalo in lactation (60 d postpartum) after slaughter. In addition, the mammary gland biopsies were conducted during the dry-off period (60 d before parturition) from other 4 buffalo by a previously described surgical procedure [21]. All tissue samples were obtained rapidly under sterile conditions and frozen instantly in liquid nitrogen.

### 2.2. Vector Construction and Small RNA Synthesis

The overexpression plasmid EGFP-*SOCS1* of buffalo *SOCS1* (accession No. XM_006079909) was constructed using pEGFP-N1 vector (Clontech Laboratories, Inc., Palo Alto, CA, USA) by PCR with a pair of specific primers containing *Xho* I and *Hind* III restriction site (forward: 5′-CTCGAGATGGTAGCACACAACCAGGT-3′; reverse: 5′-AAGCTTGATCTGGAAGGGGAAGGAGC-3′). To knock down the buffalo *SOCS1*, one pair of short hairpin RNA (shRNA) targeting buffalo *SOCS1* were designed using the software BLOCK-iT RNAi Designer (http://rnaidesigner.invitrogen.com/rnaiexpress/, accessed on 19 June 2022) (Appendix A). The shRNA was annealed and further ligated into the vector pLKO.1-TRC to construct the recombinant plasmids (pLKO.1-*SOCS1*). The obtained vectors were verified by sequencing, and then purified with EndoFree Maxi Plasmid Kit (QIAGEN, Valencia, CA, USA).

In order to investigate the regulatory effect of CEBPA on *SOCS1*, we also constructed the overexpression vector EGFP-CEBPA by the same method. The primer sequences were as follows: forward: 5′-CTCGAGATGGAGTCGGCCGACTTCTA-3′; reverse: 5′-AAGCTTCGCGCAGTTGCCCATGGCCT-3′. In addition, the small interfering RNA (siRNA) targeting *CEBPA* (siCEBPA) and negative control siRNA (siNC) were designed and further synthesized by Shanghai Sangon Biotech Company (Appendix A).

### 2.3. Cell Preparation and Treatment

The 293T cells were cultivated in the medium composed of Dulbecco’s modified Eagle medium (DMEM) (Gibco, Grand Island, NY, USA) supplemented with 10% fetal bovine serum (FBS) (Gibco) and 2% penicillin/streptomycin (Gibco), maintained at 37 °C in 5% CO_2_. The process for lentivirus generation of shRNA was performed as previously reported [22]. Briefly, when the confluence of cultured 293T cells reached 80%, pLKO.1-sh*SOCS1*, psPAX2 and pMD2.G were co-transfected into 293T cells through TransIntro™ EL Transfection Reagent (TransGen Biotech, Beijing, China) for generating lentiviral particles (sh*SOCS1*) in accordance with the manufacturer’s instructions. The 293T cells were co-transfected with pLKO.1-TRC, psPAX2 and pMD2G as the non-interfered control group (shNC). They were all measured by the serial dilution method for the infection titer of concentrated lentivirus particles.

Buffalo mammary epithelial cells (BuMECs) were isolated from the mammary gland tissue of lactating Binglangjiang buffalo (60 d postpartum) and purified based on the differential sensitivity of the cells to trypsin digestion as previously described by our group [22,23]. The BuMECs purified and identified by cytokeratin 18 (Sigma, Louis, MO, USA) were cultivated and expanded to passage five in a basal medium composed of DMEM (Gibco), 10% fetal bovine serum (Gibco), 100 μg/mL penicillin/streptomycin (Gibco) and various cytokines, including 5 μg/mL insulin (Sigma), 2 μg/mL hydrocortisone (Sigma) and 100 ng/mL epidermal growth factor (Sigma), and maintained at 37 °C under 5% CO_2_. When the cell confluence reached 80% in a 6-well cell culture plate, the cells were cultured for 24 h in a basal medium supplemented with 2 μg/mL prolactin (Sigma). Subsequently, these cells were transfected with EGFP-*SOCS1* (3 μg), EGFP-CEBPA (3 μg), siCEBPA (60 nM) and the corresponding negative controls (EGFP and siNC) according to the manufacturer’s protocol of TransIntro^TM^ EL transfection reagent (TransGen Biotech, Beijing, China). At the same time, they were transduced into sh*SOCS1* and shNC to knock down *SOCS1*. Through pre-experiments, it was determined that the best results were obtained when the cells were collected 48 h after treatment. Therefore, the cells of each treatment group were harvested 48 h later for gene and protein expression analysis.

### 2.4. Protein Subcellular Localization Analysis

The pEGFP-*SOCS1* was transfected into BuMECs for subcellular localization analysis of *SOCS1*. After 48 h of transfection, the mitochondria and nucleus of the cells were stained with Mito-Tracker Red CMXRos (Beyotime, Shanghai, China) and Hoechst 33342 (Beyotime), respectively, according to the instructions. After the staining solution was removed, observations were made with a confocal laser scanning microscope (Olympus, Tokyo, Japan).

### 2.5. Quantitative PCR (qPCR) Detection of Expression

Total RNA was extracted from transfected cells using the TRIzol (Invitrogen) and reverse transcription was performed by following the instructions of a RT reagent kit with gDNA Eraser (Takara, Dalian, China) for subsequent analysis of mRNA expression. The primers for qPCR were designed using Primer Premier 5.0 [24], and the primer sequences are listed in Appendix A. The qPCR assay was performed using the TB Green^®^ Premix Ex Taq^TM^ II (TaKaRa) on a CFX96 Real-Time System (Bio-Rad, Hercules, CA, USA). The purity of PCR products was confirmed by melting curve analysis. The efficiency of amplification was determined utilizing LinRegPCR (www.linregpcr.nl, accessed on 20 May 2022; Appendix A). The geometric mean of the Ct values of *ACTB*, *GAPDH* and *RPS23* was used as the endogenous control for mRNA expression analysis. The relative expression of mRNA was analyzed using the 2^–ΔΔCt^ model [25].

### 2.6. Protein Extraction and Western Blotting

The proteins of mammary gland tissue and transfected BuMECs were collected by trypsin digestion and lysed in RIPA buffer (Beyotime) containing 1% PMSF (Beyotime) and 1% phosphatase inhibitor cocktail (Roche, Shanghai, China). After quantifying their concentration using the BCA assay kit (Beyotime), equal amounts of protein samples (approximately 25 μg of total protein) were electrophoresed in SDS-PAGE and transferred onto nitrocellulose membranes (Millipore, Burlington, MA, USA). The membranes were probed with primary polyclonal rabbit anti-*SOCS1* (1:1000; abs131478, Absin, Shanghai, China), rabbit anti-β-casein (1:1000; bs-0466R, Bioss, Beijing, China), rabbit anti-PI3K (1:1000; bs-10657R, Bioss), rabbit anti-Phospho-PI3K (1:1000; bs-6417R, Bioss), rabbit anti-AKT1 (1:1000; bs-0115R, Bioss), rabbit anti-Phospho-AKT1 (1:1000; bs-10133R, Bioss), rabbit anti-mTOR (1:1000; bs-1992R, Bioss), rabbit anti-Phospho-mTOR (1:1000; bs-3495R, Bioss), rabbit anti-STAT5 (1:1000; bs-1142R, Bioss), rabbit anti-Phospho-STAT5 (1:1000; bs-5619R, Bioss), rabbit anti-JAK2 (1:1000; bs-23003R, Bioss), rabbit anti-Phospho-JAK2 (1:1000; #3771, Cell Signaling Technology, Danvers, MA, USA) and monoclonal mouse anti-β-actin (1:6000; HC201, TransGen Biotech, Beijing, China) at 4 °C overnight. The species reactivity of these antibodies is all for cow. Next, the membranes were further incubated with polyclonal goat anti-rabbit IgG (1:5000; #2491145, Millipore, Burlington, MA, USA) and polyclonal goat anti-mouse IgG (1:5000; #2517746, Millipore), and the immunoreactive bands were visualized using the chemiluminescent ECL Western blot detection system (Pierce, Rockford, IL, USA). Protein abundance was calculated by Alpha View SA (ProteinSimple, San Jose, CA, USA).

### 2.7. SOCS1 Promoter Cloning and Deletion Analysis

The different fragments of the *SOCS1* promoter (GenBank no. NC_059180) derived from primers that hybridize at positions −1999, −1734, −1364, −961, −614 and −77, coupled with a common downstream primer at +105 with *Xho* Ι and *Hind* III sites (Appendix A), were prepared by PCR from mixed genomic DNA, which was isolated from blood samples of 4 buffalo. They were subcloned into pGL4 vector (Promega, Madison, WI, USA) with *Xho* Ι/*Hind* III restriction enzyme to generate multiple luciferase reporter vectors (pGL-(−1999/+105), pGL-(−1734/+105), pGL-(−1364/+105), pGL-(−961/+105), pGL-(−614/+105) and pGL-(−77/+105)). The putative transcription factor binding sites were analyzed using the JASPAR database (http://jaspar.genereg.net/, accessed on 25 April 2022), AliBaba2.1 (http://gene-regulation.com/pub/programs/alibaba2/, accessed on 25 April 2022) and hTFtarget (http://bioinfo.life.hust.edu.cn/hTFtarget#!/prediction, accessed on 25 April 2022).

### 2.8. Dual-Luciferase Activity Assay

The luciferase reporter vectors containing different-length fragments of the *SOCS1* promoter were transfected into the BuMECs for luciferase expression to determine the core promoter region. Moreover, BuMECs were co-transfected with EGFP-CEBPA or siCEBPA and 5′ progressive deletion of pGL-*SOCS1* plasmids. After 48 h, luciferase assays were carried out using the Dual-Glo luciferase assay system kit (Promega). The firefly luciferase activity was normalized to the activity of Renilla luciferase (pRL-TK; Promega), and the ratio of pGL-*SOCS1* to pRL-TK was 30:1.

### 2.9. Statistical Analyses

The results were presented as mean ± SEM for each group from three independent experiments. GraphPad Prism 5 software (GraphPad Software Inc., La Jolla, CA, USA) was used for statistical analysis and visualization. Statistical significance of observed differences between the two groups was assessed using a two-tailed Student’s *t*-test, and *p* values < 0.05 were considered as significant.

## 3. Results

### 3.1. SOCS1 Differential Expression in Various Tissues

We measured the mRNA abundance of *SOCS1* in 12 tissues of lactating buffalo (Figure 1A) and observed that *SOCS1* was expressed in all examined tissues, with the highest levels in the muscle, followed by the spleen, pituitary, heart, brain and kidney. The lowest mRNA abundance among the examined tissues was found in the mammary gland. Furthermore, we also examined the differential expression of buffalo *SOCS1* mRNA and its encoded protein in the mammary gland during different physiological stages. The results showed that the mRNA (*p* < 0.01) and protein expression (*p* < 0.01) of buffalo *SOCS1* were significantly lower in lactation than that in the dry-off period (Figure 1B–D), indicating that *SOCS1* is involved in the lactation process in buffalo.

### 3.2. Nuclear–Cytosolic Localization of Buffalo SOCS1

The results of purified BuMECs identified by cytokeratin 18 are shown in Appendix A. The recombinant vector of EGFP-*SOCS1* was transfected into BuMECs to determine the distribution of buffalo *SOCS1*. The results of transfection showed that most of the green fluorescent protein (GFP) of EGFP-*SOCS1* coincided with the red fluorescence of the mitochondria; some of the GFP overlapped with the blue fluorescence of the nucleus (Figure 2). The above results suggest that the *SOCS1* protein is mainly distributed in the cytoplasm of BuMECs, but some are also distributed in the nucleus.

### 3.3. SOCS1 Suppresses the Milk Protein Synthesis in BuMECs

To understand the impact of *SOCS1* on milk protein synthesis, EGFP-*SOCS1* or sh*SOCS1* were transfected into BuMECs. Compared with the negative controls, the expression level of *SOCS1* in BuMECs increased 106-fold after treatment with EGFP-*SOCS1* (*p* < 0.05), while the expression level of *SOCS1* in cells treated with sh*SOCS1* decreased by 67.6% (*p* < 0.05) (Figure 3A,B). For the effect of *SOCS1* on milk protein synthesis, we examined the expression changes of *CSN2* and *CSN3* in BuMECs. Compared with the negative controls, EGFP-*SOCS1* markedly reduced the mRNA abundance of *CSN2* (*p* < 0.01) and *CSN3* (*p* < 0.01) (Figure 3C), whereas sh*SOCS1* significantly increased the expression of *CSN2* (*p* < 0.01) and *CSN3* (*p* < 0.01) (Figure 3D). CSN2 (β-casein) is the second most abundant protein in cow’s milk and can be employed as an indicator of milk protein synthesis in mammary epithelial cells [26]. Thus, after analyzing the expression of *CSN2* at the mRNA level, we also examined the protein level of *CSN2* to confirm the regulatory effect of *SOCS1* on the milk protein synthesis. The results displayed that the expression of CSN2 protein in the group treated with EGFP-*SOCS1* decreased significantly compared to that in the EGFP group (*p* < 0.01). On the contrary, the expression of CSN2 was significantly up-regulated in the group treated with sh*SOCS1* (*p* < 0.05) (Figure 3E,F).

### 3.4. SOCS1 Inhibits the Activation of the mTOR and JAK2-STAT5 Pathways

To further determine the function of *SOCS1* in milk protein synthesis, the expression changes of genes related to the mTOR and JAK2-STAT5 pathways were investigated when *SOCS1* was overexpressed and knocked down in BuMECs. We examined the changes in the expression of central genes, including genes in the upstream pathway (*PI3K*, *AKT1*, *mTOR*) and genes in the downstream pathway (*4EBP1*, *EIF4E*, *S6K1*) of mTOR signaling, as well as genes (*PRLR*, *JAK2*, *STAT5a*, *STAT5b*, *ELF5*) in JAK2-STAT5 signaling. The results showed that the *SOCS1* overexpression significantly decreased the mRNA abundance of *PI3K* (*p* < 0.01), *mTOR* (*p* < 0.01) and *EIF4E* (*p* < 0.01) in BuMECs (Figure 4A,B). In contrast, the treatment with sh*SOCS1* significantly increased the mRNA expression of *PI3K* (*p* < 0.05), *AKT1* (*p* < 0.01), *mTOR* (*p* < 0.05) and *EIF4E* (*p* < 0.05) in mTOR signaling pathways (Figure 4D,E). However, the mRNA expression of *4EBP1* and *S6K1* (*p* > 0.05) did not change after transfection with EGFP-*SOCS1* or sh*SOCS1* in BuMECs. Moreover, the mRNA expression of genes involved in the JAK2-STAT5 pathway, *PRLR* (*p* < 0.001), *STAT5a* (*p* < 0.05), *STAT5b* (*p* < 0.01) and *ELF5* (*p* < 0.05) were significantly decreased upon the up-regulation of *SOCS1* (Figure 4C). The down-regulation of *SOCS1* caused significantly increased expression of *PRLR* (*p* < 0.01), *JAK2* (*p* < 0.01), *STAT5b* (*p* < 0.01) and *ELF5* (*p* < 0.05) (Figure 4F). Importantly, Western blot detection showed that *SOCS1* overexpression decreased the protein levels of PI3K, p-PI3K, p-AKT1, mTOR, p-mTOR, p-JAK2, STAT5 and p-STAT5 in BuMECs, whereas *SOCS1* knockdown had the opposite effects (Figure 4G–J). However, the protein levels of AKT1 and JAK2 remained almost unchanged. These data revealed that *SOCS1* affects milk protein synthesis through the inhibition of the mTOR and JAK2-STAT5 signaling pathways.

### 3.5. CEBPA Is Required for Induction of SOCS1 Expression

To investigate the role of CEBPA on the expression of *SOCS1*, the changes in promoter activity, mRNA and protein expression of *SOCS1* when *CEBPA* was overexpressed or knocked down were analyzed. It was found that *CEBPA* overexpression dramatically increased the luciferase activity of the pGL-(−1999/+105) (*p* < 0.01), whereas the luciferase level of the pGL-(−1999/+105) transfected with siCEBPA was significantly inhibited relative to siNC (*p* < 0.01) (Figure 5A). The *SOCS1* mRNA expression was up-regulated by *CEBPA* overexpression (*p* < 0.01), but its expression was markedly down-regulated by *CEBPA* inhibition (*p* < 0.05) (Figure 5B). In addition, the protein expression level of *SOCS1* was consistent with the trend of mRNA expression after *CEBPA* overexpression or knockdown in BuMECs (Figure 5C,D).

### 3.6. Identification of the Core Promoter Region of Buffalo SOCS1

To determine the core promoter region of buffalo *SOCS1*, pGL-*SOCS1* plasmids with 5′ progressive deletions were constructed (Appendix A) and transfected directly into BuMECs for the expression of luciferase. Luciferase detection showed that compared with other deletion constructs, the pGL-(−1364/+105) had the highest luciferase activity (Figure 6A). The deletion from −1999 bp to −1364 bp showed a significant increase in luciferase activity, indicating that this region may contain some negative regulatory elements. The deletion from −1364 bp to −961 bp showed a dramatic decrease in luciferase activity (*p* < 0.05), suggesting that some important positive regulatory elements were deleted. When the deletion reached −614 bp, we found that the luciferase activity was significantly reduced (*p* < 0.01). Subsequently, with the deletion of the 5′ flanking region, the activity continued to decrease until it was similar to that of the empty vector pGL4 (*p* < 0.05). The above results reveal that the promoter region ranging from −1364 to +105 is the proximal core promoter region of buffalo *SOCS1*. Analysis of transcription factor response elements in the core promoter region demonstrated that there are two CEBPA (from −1346 to −1336 and −1085 to −1076) and two NF-κB binding sites (from −931 to −921 and −672 to −662) in this region (Figure 6B).

### 3.7. CEBPA and NF-κB Binding Sites Are Responsible for Induction of SOCS1 Promoter Activity by CEBPA

To identify which *cis*-regulatory elements are responsible for CEBPA-mediated *SOCS1* regulation, we assessed the effect of CEBPA on the promoter activity of *SOCS1* with different lengths in BuMECs. Compared with the control (EGFP), the overexpression of *CEBPA* effectively enhanced the activity of promoters containing CEBPA and NF-κB binding sites (Figure 7A). Meanwhile, its knockdown significantly reduced the luciferase activity of recombinant vectors containing these *cis*-regulatory elements (Figure 7B). However, CEBPA had no effect on the activity of promoters that did not contain CEBPA and NF-κB binding sites. Collectively, these results indicated that CEBPA promotes *SOCS1* transcription via the CEBPA and NF-κB binding sites located in the *SOCS1* promoter in BuMECs.

## 4. Discussion

Lactation is a complex and dynamic biological process that is an important part of the mammalian reproductive process [27]. Lactation traits are extremely important economic traits in dairy production, which include milk protein content, milk protein rate, milk fat content, milk fat rate and milk yield, and so on. Milk protein is an important source of dietary protein for humans, and its content is a key economic indicator used to evaluate milk quality and process characteristics [28,29,30]. *SOCS1* has been revealed to be involved in the regulation of milk protein synthesis in mice, but data on it in buffalo are extremely scarce. In the present study, it was found that *SOCS1* was highly expressed in the muscle and spleen of buffalo. In addition, this gene was moderately expressed in the lung, which is also consistent with the findings in pigs [31]. Cytokines play a crucial role in mammary gland development, which is reflected by their different expression in the mammary gland at various physiological periods [32]. As a cytokine signaling suppressor, *SOCS1* expression has been reported to decrease from pregnancy to lactation in the mammary gland of cows [8]. Consistent with this, both mRNA and protein expression levels of *SOCS1* in the mammary gland of buffalo during lactation in this study were markedly lower than that in the dry-off period, suggesting that *SOCS1* is closely associated with buffalo mammary gland development and lactation.

The inhibition of JAK2 by *SOCS1* is achieved by blocking substrate access to the JAK2 kinase via a fragment of 24 amino acids called the kinase inhibition region. *SOCS1* recruits phosphorylated Tyr-1007 in the activation loop of JAK2 through its SH2 domain, thereby inhibiting the catalytic activity of JAK2 [33]. Meanwhile, *SOCS1* can directly bind and target phosphorylated JAK2 for degradation through its E3 ubiquitin-ligase-like activity [34]. These processes take place in the cytoplasm of the cell. In addition, *SOCS1* has been reported to have a nuclear localization signal, and recent studies have shown that it is detected in the nucleus, where it interacts with nuclear factor-κB (NF-κB), and is also part of the DNA damage response [35,36]. Our data show that buffalo *SOCS1* is mainly distributed in the cytoplasm and partially in the nucleus of BuMECs, which is consistent with previous findings, revealing that *SOCS1* interacts with JAK2 in the cytoplasm and also functions in the nucleus of BuMECs.

Previous studies have shown that *SOCS1* is a key physiological attenuator of PRLR signaling, suggesting that *SOCS1* can inhibit the expression or activity of PRLR and its downstream proteins [9]. In human 293 cells, the overexpression of *SOCS1* resulted in a significant decrease in PRLR-mediated phosphorylation of JAK2 and STAT5, and eliminated the ability of PRLR to induce activation of the *CSN2* promoter [37]. Consistently, the overexpression of *SOCS1* here reduced the mRNA and protein abundance of CSN2, accompanied by the decreased phosphorylation of JAK2 and STAT5, while the knockdown of this gene had the opposite effect in BuMECs. It is worth noting that ELF5 also plays a role in activating STAT5, and its encoding gene is the target gene of STAT5 [38]. In this experiment, the overexpression of *SOCS1* decreased the expression of *ELF5*, whereas the knockdown of this gene increased the expression of *ELF5.* We speculate that since *SOCS1* altered the expression and phosphorylation of STAT5 protein, thereby indirectly regulating *ELF5* expression in BuMECs. Intriguingly, we have further confirmed that *SOCS1* can affect the protein expression levels of PI3K, p-PI3K, p-AKT1, mTOR and p-mTOR. Evidence suggests that phosphorylation of insulin receptor substrate 1 (IRS1) activates the PI3K-AKT-mTOR signaling pathway, and SOCS proteins are important mediators of IRS1 degradation, and target IRS1 through interaction with SOCS box motif [39,40]. Therefore, we speculate that buffalo *SOCS1* may affect the PI3K-AKT-mTOR signaling pathway through IRS1. Overall, *SOCS1* inhibits the synthesis of buffalo milk proteins by affecting the mTOR and JAK2-STAT5 pathways.

A previous study confirmed that STAT5 induces *CEBPA* expression during basophil and mast cell development [41]. As an essential transcription factor, CEBPA takes part in many biological processes, such as negative regulation of cell cycle progression, cell differentiation and apoptosis [42]. In patients with acute myeloid leukemia, *SOCS1* expression is negatively correlated with mutant CEBPA [43]. It has also been found that CEBPA indirectly induces the expression of *SOCS1* through miR-29b [18]. These findings suggest that CEBPA is closely associated with the expression of *SOCS1*. To date, whether CEBPA directly regulates the expression of buffalo *SOCS1* and its regulatory mechanism remain unclear. The results in this study showed that the overexpression of *CEBPA* led to increased activity of *SOCS1* promoter, while the knockdown of *CEBPA* had the opposite effect. Furthermore, the mRNA and protein abundance of *SOCS1* varied with *CEBPA* expression, suggesting that CEBPA can regulate the expression of buffalo *SOCS1*. This promoter deletion experiment demonstrated that the overexpression of *CEBPA* enhanced the activity of *SOCS1* promoter fragments representing the promoter regions between −1734/+105, −1364/+105 and −961/+105 bp. Sequence analysis of the *SOCS1* promoter identified two potential CEBPA binding sites in the promoter regions between −1734/+105 and −1364/+105. Therefore, CEBPA can bind to these sites and thus promote the *SOCS1* expression. Notably, although the fragment of *SOCS1* promoter from −961 to +105 bp did not contain the CEBPA binding site, up-regulation of *CEBPA* still enhanced its activity, suggesting that CEBPA indirectly regulates the activity of *SOCS1* promoter. We identified two NF-κB binding sites in this fragment of the promoter. As proposed by Paz-Priel et al., CEBPA can induce NF-κB expression by binding to its promoter [44]. In addition, *SOCS1* may be induced by NF-κB in mice [45]. Therefore, it is suggested that CEBPA can also indirectly regulate the expression of *SOCS1* in an NF-κB-dependent manner. This was further confirmed here by the knockdown experiment of *CEBPA* in BuMECs. This work reveals that buffalo *SOCS1* is a target gene of CEBPA, and that the binding sites of CEBPA and NF-κB are essential elements for CEBPA-mediated transcriptional regulation of *SOCS1*.

## 5. Conclusions

Our results suggest that buffalo *SOCS1* can inhibit milk protein synthesis through the mTOR and JAK2-STAT5 pathways in BuMECs. The promoter region of *SOCS1* ranged from −1364 to +105 contains CEBPA and NF-κB binding sites, and was identified as the proximal core promoter. CEBPA regulates the transcription of buffalo *SOCS1* through CEBPA and NF-κB binding sites in the *SOCS1* promoter. This study can provide a basis for elucidating the genetic basis and regulatory mechanism of buffalo milk protein traits.

## Figures and Tables

**Figure 1 foods-12-00708-f001:**
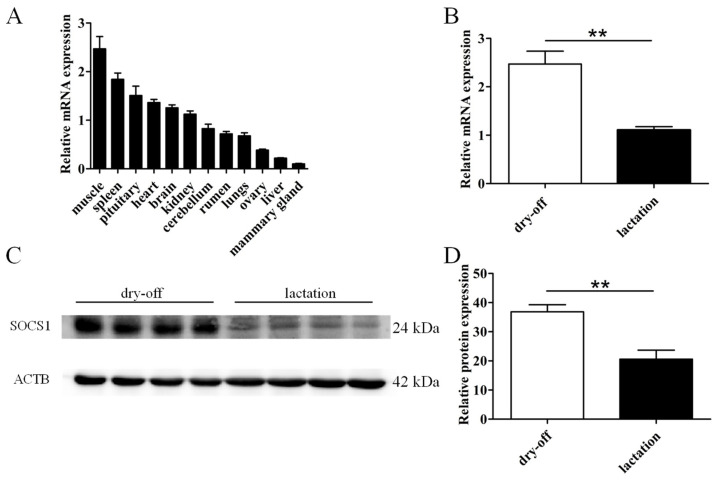
Differential expression of *SOCS1* in various tissues of buffalo. (**A**) Differential mRNA expression of *SOCS1* in 12 buffalo tissues. (**B**) Differential mRNA expression of buffalo *SOCS1* in the mammary gland between lactation and dry-off period. (**C**,**D**) Differential expression of buffalo *SOCS1* protein in the mammary gland between lactation and dry-off period. Data are presented as means ± SEM for three individual cultures; asterisks indicate differences between dry-off and lactation: ** *p* < 0.01.

**Figure 2 foods-12-00708-f002:**
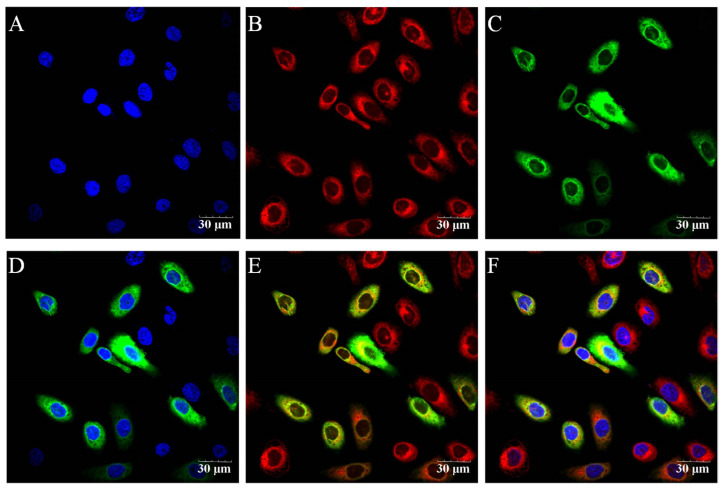
Immunofluorescent staining of nucleus (blue), mitochondria (red) or EGFP-*SOCS1* (green) and confocal microscopy analysis in BuMECs. (**A**,**B**) Nucleus and mitochondria stained with Hoechst 33342 and Mito-Tracker, respectively; (**C**) green fluorescent protein was expressed by EGFP-*SOCS1*; (**D**) merged overlaid nucleus and EGFP-*SOCS1*; (**E**) merged overlaid mitochondria and EGFP-*SOCS1*; (**F**) merged overlaid nucleus, mitochondria and EGFP-*SOCS1*.

**Figure 3 foods-12-00708-f003:**
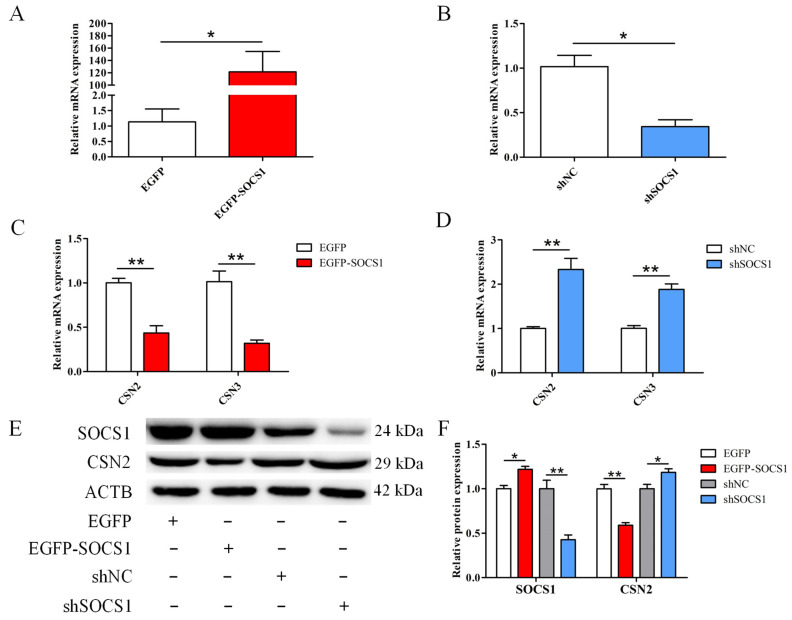
Effect of *SOCS1* on milk protein synthesis in buffalo mammary epithelial cells (BuMECs). (**A**,**B**) The expressions of *SOCS1* in BuMECs treated with the EGFP-*SOCS1*, sh*SOCS1* and corresponding negative controls (EGFP and shNC). (**C**,**D**) Effects of EGFP-*SOCS1* or sh*SOCS1* on the expression of *CSN2* and *CSN3*. (**E**,**F**) Effects of EGFP-*SOCS1* or sh*SOCS1* on the expression level of CSN2 protein. Data are presented as means ± SEM for three individual cultures; * *p* < 0.05, ** *p* < 0.01.

**Figure 4 foods-12-00708-f004:**
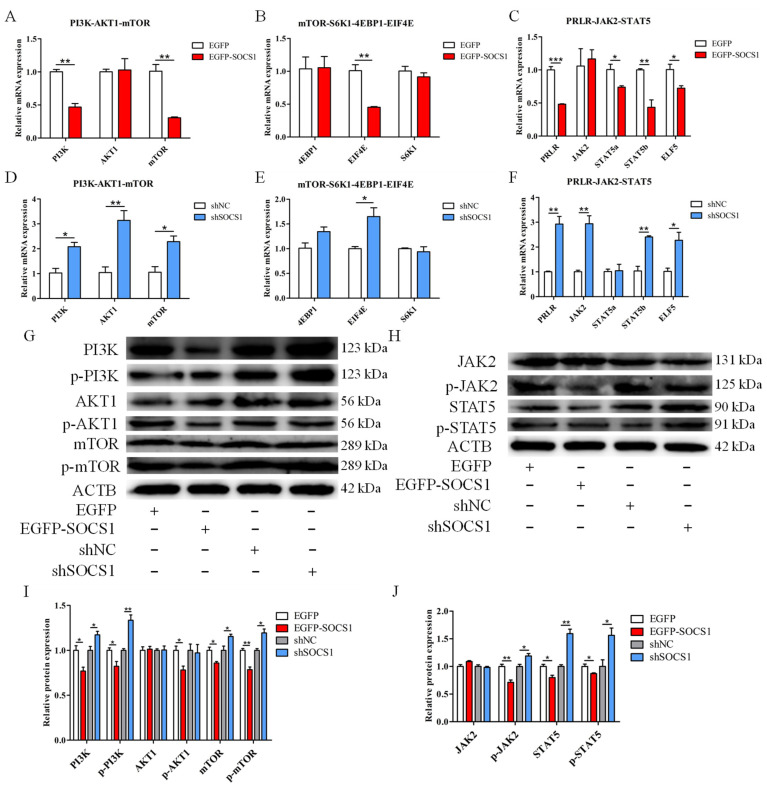
Effect of *SOCS1* on the expression of genes and proteins related to the mTOR and JAK2-STAT5 pathways in BuMECs. (**A**,**D**) Effects of EGFP-*SOCS1* or sh*SOCS1* on the expression of genes related to the upstream pathway of mTOR signaling. (**B**,**E**) Effects of EGFP-*SOCS1* or sh*SOCS1* on the expression of genes related to the downstream pathway of mTOR signaling. (**C**,**F**) Effects of EGFP-*SOCS1* or sh*SOCS1* on the expression of genes participating in the JAK2-STAT5 signaling. (**G**–**J**) Effects of EGFP-*SOCS1* or sh*SOCS1* on the expression of proteins related to the mTOR and JAK2-STAT5 pathways. Data are presented as means ± SEM for three individual cultures; * *p* < 0.05, ** *p* < 0.01, *** *p* < 0.001.

**Figure 5 foods-12-00708-f005:**
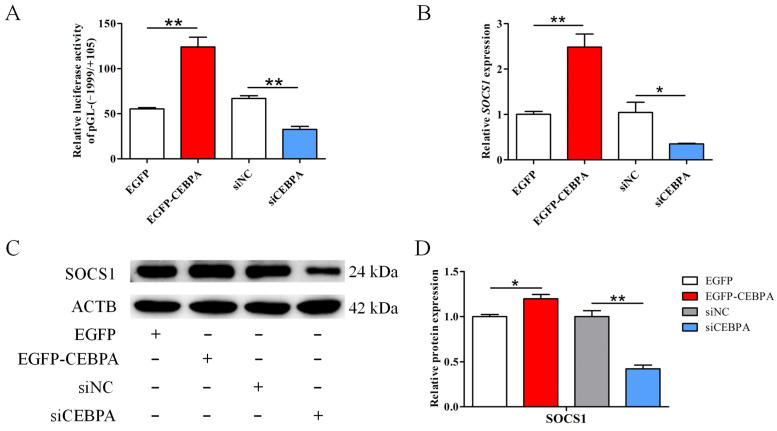
CEBPA promotes the transcription and expression of *SOCS1* in buffalo mammary epithelial cells (BuMECs). (**A**) The BuMECs were co-transfected with the vector pGL-(−1999/+105) of *SOCS1* promoter and either the EGFP-CEBPA or EGFP vector, or either the siCEBPA or siNC. (**B**–**D**) The mRNA and protein expression of *SOCS1* in BuMECs following transfection with EGFP-CEBPA or siCEBPA, as well as their corresponding negative controls. Data are presented as means ± SEM for three individual cultures; * *p* < 0.05, ** *p* < 0.01.

**Figure 6 foods-12-00708-f006:**
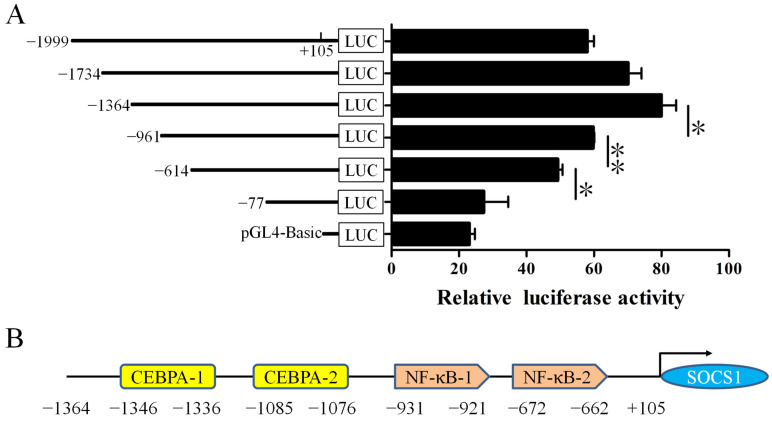
Identification of the core promoter region of *SOCS1* and analysis of transcription factor binding sites. (**A**) Relative luciferase activity (Firefly: Renilla) after 48 h transfection with 5′ progressive deletions of pGL-*SOCS1* plasmids (pGL-(−1999/+105), pGL-(−1734/+105), pGL-(−1364/+105), pGL-(−961/+105), pGL-(−614/+105) and pGL-(−77/+105)). (**B**) Predicted transcription factor binding sites in the core promoter region of buffalo *SOCS1*. Data are presented as means ± SEM for three individual cultures; * *p* < 0.05, ** *p* < 0.01.

**Figure 7 foods-12-00708-f007:**
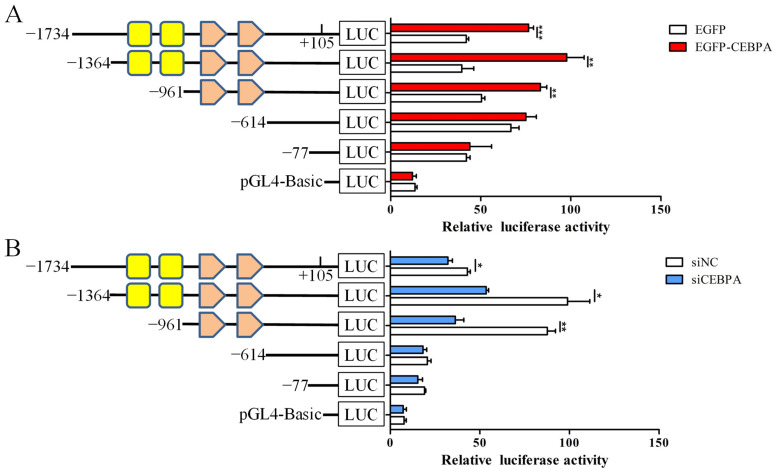
Identification of the *cis*-regulatory elements of CEBPA regulating the transcriptional activity of buffalo *SOCS1* in BuMECs. (**A**) The effect of *CEBPA* overexpression on the activity of *SOCS1* promoters of different lengths. (**B**) The effect of *CEBPA* knockdown on the activity of *SOCS1* promoters of different lengths. Yellow rectangles represent CEBPA binding sites and orange pentagons represent NF-κB binding sites. Data are presented as means ± SEM for three individual cultures; * *p* < 0.05, ** *p* < 0.01, *** *p* < 0.001.

## Data Availability

The data analyzed during the current study are available from the corresponding author on reasonable request.

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
