# Peer review of "CEBPA-Regulated Expression of SOCS1 Suppresses Milk Protein Synthesis through mTOR and JAK2-STAT5 Signaling Pathways in Buffalo Mammary Epithelial Cells"

_foods, 2023, doi:10.3390/foods12040708_

Round 1

Reviewer 1 Report

In this study Fan, et al. suggested that buffalo SOCS1 can inhibit milk protein synthesis through the mTOR and JAK2-STAT5 pathways in BuMECs and provide the genetic basis and regulatory mechanism for buffalo milk protein traits.

This manuscript is well-written with logic and seems to be interesting to the audience. Just minor spelling mistakes that need to check.

Reviewer 2 Report

The article “SOCS1 suppresses milk protein synthesis through mTOR and JAK2-STAT5 signaling pathways and its expression is directly regulated by CEBPA in buffalo mammary epithelial cells” by Fan et al., evaluated the effect of SOCS1 on inhibition of milk protein synthesis and its expression is directly regulated by CEBPA in buffalo mammary epithelial cells. Authors carried out a series of experiments to show that buffalo SOCS1 plays a significant role in affecting milk protein synthesis through mTOR and JAK2-STAT5 signaling pathways, and its expression is directly regulated by CEBPA. This manuscript aims to present a very interesting and well-conceived and developed research. The authors, starting from their aim, explored this interesting topic in a very deep and multimethodology way, which allows the obtention of soundly based results that truly add new knowledge in the field of ruminant lactation physiology. For these reasons, in my opinion the article needs just some minor revisions before to be considered suitable for the publication in the journal.

1.     I wonder if the protein content in milk is different in different lactation stages, if so, please explain why author collected samples from buffalo at 60 days postpartum.

2.     In Fig 1A, I suggest author to remove the letters above the bar.

3.     P-value, capital letters throughout the manuscript.

4.     Please add the molecular weight of protein.

5.     Finally, English should be revised by a native speaker, since some flaws are present throughout the manuscript.

Reviewer 3 Report

I have reviewed the manuscript »SOCS1 suppresses milk protein synthesis through mTOR and 2 JAK2-STAT5 signaling pathways and its expression is directly regulated by CEBPA in buffalo mammary epithelial cells«. The manuscript contains some relevant data, however I have the following concerns:

MAJOR

-          Consider changing the title of the manuscript (e.g. CEBPA regulated expression of SOCS1 suppresses milk protein synthesis through mTOR and  JAK2-STAT5 signaling pathways in buffalo mammary epithelial cells).

-          Please explain briefly in the introduction how hormonal regulation of lactation associates with the mentioned transcription factors.

-          The work includes animal experiment- biopsies were performed. It is necessary to include ethics approval from corresponding animal ethics research committee.

-          The methods section should be described more clearly and in more detail (see also comments below).

-          ELISA should be performed to quantitate expression changes of casein genes are translated to the protein level; the same goes for the selected markers of mTOR and JAK/STAT pathways, and CEBPA. I would not consider WB signals a reliable quantitative method.

-          The quality of written English needs to be improved (especially in the discussion sections).Discussion should be rewritten and presented in a more cohesive way. There are some conclusions made on speculations (without presenting clear evidence), which should be alleviated (for example, it is difficult to make conclusions about tissue-specific gene functions based merely on gene expression comparison in different tissues)

MINOR

-          Line 10: »milk protein…« content?

-          Line 14: italicize SOCS1

-          Line 31: »The synthesis and secretion of milk is the major and important…« (consider changing the sentence: e.g. The synthesis and secretion of milk is the major and the most important…

-          Line 33: you may delete »activities«

-          Line 37: protein -> proteins

-          Line 74: »between it and« -> with

-          Line 79: »will« -> aims to

-          Which reagent was used for transfecting plasmids into HEKs and BuMECs? It is not clear when transfection of plasmids was used and when lentiviral vectors were used. Also, I think that when you used lentiviral vectors term transduction should be used.

-          Line 144: I think that »fluorescence« is not necessary as it is generally known that the detection relies on fluorescent dies

-          Line 148: »of« -> for

-          Line 155: reference for ddCt method is missing

-          In the methods you list antibodies for WB, but it is not clear against which species antigen are they produced (for example, polyclonal rabbit-anti SOCS1 is produced in rabbit, but it is unclear what is its species reactivity – is it produced against buffalo SOCS1?...). Furthermore, you describe that »the membranes were further incubated with appropriate secondary antibodies« - what are appropriate secondary antibodies, be more specific.

-          Line 204 :difference -> differential

-          Figure S1: ordinary images?, the second photo in the top row is of bad quality

-          Line 222: »The results of purified BuMECs identified by cytokeratin 18 are shown in Supplementary Figure S1.« What do you mean by results of purified BuMECS? It is an image of CK18 staining.

-          It is not explained why did you checked for co-localization of SOCS1 and mitochondria?

-          Line 243: »detected« -> examined

-          Line 246-248: That is not true for ruminants in general, it depends on the species.

-          I am not sure what is shown on figures 3F, 4I and 4J. Is that protein content assessed from WB signal?

-          How do you explain deletion from -2000 to -1364 bp in the SOCS1 promoter had positive effect on luciferase expression?

-          Line 358: »in the production of lactating livestock« -> in dairy production

-          Line 359-362: »Among them, milk protein is an important source of dietary protein for humans, and its content is a key economic indicator to evaluate milk quality and process characteristics of dairy cows [27-29].« Please rephrase this sentence (for example, delete »among them«, milk protein is one of the key indicators…, delete »dairy cows«).

-          If you write gene name in italics you shouldn't add word »gene« after the symbol (e.g. »SOCS1 gene« line 364).

Best regards
